# Is Benin on track to reach universal household coverage of basic water, sanitation and hygiene services by 2030?

Nicolas Gaffan[1]*, Cyriaque Dégbey[2,3], Alphonse Kpozèhouen[1], Yolaine Glèlè Ahanhanzo[1], Roch Christian Johnson[4], Roger Salamon[5]

1 Department of Epidemiology and Biostatistics, Regional Institute of Public Health, University of Abomey-Calavi, Ouidah, Benin, 2 Department of Environmental Health, Regional Institute of Public Health, University of Abomey Calavi, Ouidah, Benin, 3 University Hospital Hygiene Clinic, National Hospital and University Centre Hubert Koutoukou Maga, Cotonou, Benin, 4 Inter-Faculty Environmental Research Center for Sustainable Development, University of Abomey-Calavi, Abomey-Calavi, Benin, 5 Institute of Public Health, Epidemiology and Development, Victor Segalen University, Bordeaux, France

* gafnicolas@gmail.com

## Abstract

### Introduction

With the end of the Millennium Agenda, the United Nations Member States adopted the Sustainable Development Agenda in 2015. This new agenda identifies 17 Sustainable Development Goals (SDGs) and 169 targets for 2030, including Water, Sanitation and Hygiene (WASH).

### Objective

To study the evolution of household access to WASH services over the last two decades in Benin and make projections for 2030.

### Methods

In this study, secondary analyses were performed using the datasets of the Demographic and Health Surveys in Benin from 2001 to 2017–2018. The statistical unit was the household. The achievement of the WASH SDGs targets was monitored through the proportion of households using individual basic WASH services, the proportion of households using surface water for drinking, and the proportion of households practising open defecation. The study generated Annual Percentage Changes (APCs) for outcome variables. Based on the APCs between 2001 and 2017–2018, projections were made for 2030.

### Results

From 2001 to 2017–2018, household access to individual basic WASH services increased from 50.54% to 63.98% (APC = +1.44%), 5.39% to 13.29% (APC = +5.62%), and 2.12% to 10.11% (APC = +9.92%), respectively. At the same time, the prevalence of surface water consumption and open defecation among households decreased from 10.54% to

**Data availability statement:** Data supporting the results of this study are available without restriction on the DHS program website (https://dhsprogram.com/data/available-data-sets.cfm).

**Funding:** The author(s) received no specific funding for this work.

**Competing interests:** The authors have declared that no competing interests exist.

5.84% (APC = -3.52%) and 67.03% to 53.91% (APC = -1.31%), respectively. If the trend observed between 2001 and 2017–2018 remains unchanged, the national coverage of households with basic individual WASH services would be 76.50%, 26.33% and 10.51%, respectively, by 2030. The prevalence of surface water consumption and open defecation among households would be 3.73% and 45.71%, respectively, by 2030.

## Conclusion

Benin achieved significant progress in household coverage of adequate WASH services over the last two decades. However, progress appears insufficient to achieve universal coverage of households with basic WASH services, and eliminate surface water consumption and open defecation by 2030. There is a need to strengthen research into the drivers of household access to adequate WASH services.

## Introduction

In September 2000, 192 members of the United Nations (UN), including Benin, and a set of international organisations adopted the Millennium Development Goals (MDGs) at the Millennium Summit [1]. Eight in number, the MDGs called on states "to combat poverty, hunger, disease, illiteracy, environmental degradation and discrimination against women by 2015" [2, 3]. These goals, broken down into targets, provided a global action plan for development [2]. The focus of target 7c was water and sanitation [2]. It called on countries to "halve the percentage of people without sustainable access to safe drinking water and basic sanitation by 2015" (as compared to the 1990 level) [4]. The progress towards this target was monitored through the proportion of the population using improved drinking water and sanitation facilities [4]. Improved drinking water sources are water points protected from external contamination. Improved sanitation facilities are unshared, ensure hygienic separation of human excreta and avoid contact with people [4, 5]. We emphasize that in this document, "access" and "coverage" have been considered synonymous terms, referring to the primary utilization of a given Water, Sanitation and Hygiene (WASH) service by households.

During the MDGs era, globally, the use of improved drinking water facilities increased from 76% in 1990 to 91% in 2015 [5]. At the same time, coverage with improved sanitation facilities increased globally from 54% in 1990 to 68% in 2015 [5]. Yet, the global target was 77%. There is a shortfall of nine percentage points or nearly 700 million people who have not been reached [5]. Behind the global figures, significant disparities remained. Central Asia, Northern Africa, Oceania and Sub-Saharan Africa did not meet the MDG water target, although they did at the global level [5]. In addition, regions other than Sub-Saharan Africa, Oceania and South Asia were able to halve the population without access to improved sanitation, in contrast to the global situation [5]. In total, 147 countries met the MDG target for water, 95 countries met the target for sanitation, and 77 countries met the target for both water and sanitation [5]. In Benin, the MDG target for water was reached (78% in 2015 against 57% in 1990). However, this was not achieved for the MDG target for sanitation (20% in 2015 against 7% in 1990, for a target of 53%) [5].

With the end of the Millennium Agenda, the Sustainable Development Agenda was adopted in September 2015 [6]. This new Agenda identifies 17 Sustainable Development Goals (SDGs) and 169 targets for 2030 [6]. The difference between the MDGs and the SDGs

lies mainly in the issue of sustainability [7]. While the MDGs dealt mainly with social issues, the SDGs consider "the dimensions of sustainable development: economic growth, social inclusion and environmental protection" [7]. In the new post-2015 schedule, water and sanitation have gained more prominence and now appear as one of the 17 goals [6, 7]. Goal 6 is dedicated to water and sanitation issues and is broken down into eight targets (in particular 6.1 and 6.2) to raise the quality of water and sanitation services [6, 7]. Besides, the hygiene component received special attention through target 6.2 [6, 8]. The inclusion of hygiene in SDGs target 6.2 reflects a greater awareness of its relevance [8]. Hygiene refers to many behaviours, including hand hygiene, menstrual hygiene and food hygiene [8]. In particular, hand hygiene has been identified as a priority and considered an appropriate indicator for national and global monitoring [8]. In many developing countries, the priority remains to ensure universal access to at least a basic level of service [8]. This unfinished target of the MDGs period has remained a central focus of SDG 1, which includes a target to ensure universal access to basic services, especially for the poor and vulnerable [6, 8].

Nevertheless, in 2020, 489 million people (7%) worldwide still lacked access to improved drinking water services, including 122 million people (2%) using surface water for drinking [9]. Also, 494 million people (6%) practice open defecation [9]. About 670 million people (29%) have no means (water and soap) for hand hygiene [9]. Besides, estimates in 2020 showed that the world was on track to meet the 2030 WASH targets, with Sub-Saharan Africa and Oceania most off track [9]. This issue is particularly alarming as the COVID-19 pandemic has highlighted the need for enhanced political commitment and accountability to close the WASH gap in health facilities and the general population [10]. In Benin, a study on 2017–2018 national data indicated that 64%, 13% and 10% of households had access to individual basic WASH services, respectively [11]. Besides, in 6% and 54% of households, members consumed surface water as drinking water and practised open defecation, respectively [11]. Data from 2020, five years after the adoption of the SDGs, indicated that 65%, 17% and 12% of Beninese had access to basic drinking water, sanitation and hygiene facilities, respectively [9]. According to the same report, the proportion of the population consuming surface water and practising open defecation was 3% and 52%, respectively [9]. In these conditions, is Benin on track to achieve universal WASH coverage by 2030?

This study examines the evolution of household access to WASH services over the last two decades and makes projections for 2030.

## Methods

### Study setting

Benin is a West African state covering an area of 114,763 km$^2$ [12]. Benin has 12 departments: Alibori, Atacora, Atlantique, Borgou, Collines, Couffo, Donga, Littoral, Mono, Ouémé, Plateau and Zou since the law n°97–028 of 15 January 1999 on the organisation of the territorial administration [13]. The population recorded in 2002 was 6,769,914 inhabitants [14]. The fourth General Census of Population and Housing (RGPH-IV) in 2013 counted 10,008,749 inhabitants, giving an annual growth rate of 3.5% over 2002–2013, slightly higher than that obtained over 1992–2002 (3.25%) [15]. Projections for 2019 estimated Benin's population at 11,884,127 inhabitants [12]. The sex ratio increased from 94.3 men per 100 women in 2002 to 96.5 men per 100 women in 2013 and 96.8 men per 100 women in 2019 [12, 14].

### Study type and data source

In this study, secondary analyses were performed using the Demographic and Health Survey (DHS) datasets in Benin. They were downloaded after a request via the website https://

dhsprogram.com/. The DHSs are a series of surveys carried out since 1984 in Africa, Latin America, Central America, the Caribbean, Asia, Europe and the Near East as part of the DHS program [16]. They aim to produce demographic and health indicators from nationally representative samples of women aged 15–49, men aged 15–64 and children under five [16]. In Benin, the DHSs are conducted by the National Institute of Statistics and Demography (formerly the National Institute of Statistics and Economic Analysis) in collaboration with the Ministry of Health and with the support of technical and financial partners. In total, Benin has conducted five DHSs: DHS-I in 1996, DHS-II in 2001, DHS-III in 2006, DHS-IV in 2011–2012 and DHS-V in 2017–2018. Details on the DHS Program are described elsewhere [16].

## Study population

The statistical unit was the household. The study population consisted of the successfully investigated households in the following surveys: DHS-II in 2001, DHS-III in 2006, DHS-IV in 2011–2012 and DHS-V in 2017–2018. Data from DHS-I in 1996 were not included due to the absence of variables to assess the hygiene component and differences in variables related to water and sanitation with subsequent surveys.

## Sampling

In Benin, the DHS-II, DHS-III, DHS-IV and DHS-V were based on a nationally representative sample of the Beninese population obtained through a two-stage stratified random sampling. The departments were stratified into urban and rural, except for the Littoral, an entirely urban stratum. From DHS-III, 23 sampling strata were created. In the DHS-II, only 13 strata were considered due to the previous administrative division, which consisted of six departments (Atacora, Atlantique, Borgou, Mono, Ouémé, Zou). In DHS-II, Atacora was defined as Atacora and Donga; Atlantique as Atlantique and Littoral; Borgou as Borgou and Alibori; Mono as Mono and Couffo; Ouémé as Ouémé and Plateau and Zou as Zou and Collines. Then, in each stratum, a specific number of Primary Sample Units (PSUs) were selected, using a systematic drawn with probability proportional to size (1st stage). The list of Enumeration Areas (EAs) established during the previous population census surveys served as the sampling frame. After listing the households within the selected PSUs, a systematic sample (equal probability) of households was drawn from each one. The full-text reports of the DHS-II, DHS-III, DHS-IV and DHS-V present full details of the sampling design [17–20]. Table 1 synthesizes the sampling design features of these surveys.

## Study variables

SDG1 calls on UN member states to "eradicate poverty in all its forms" and includes a target to "ensure universal access to basic services, especially for the poor and vulnerable" [6]. At this level, three indicators were considered: the proportion of households using basic drinking water services, the proportion of households using basic sanitation services, and the proportion of households using basic hygiene services (Table 2) [6]. For SDG6, the proposed global indicators for targets 6.1 and 6.2 were: the proportion of households using safely managed water services and the proportion of households using safely managed sanitation facilities [6]. The study did not include these two indicators due to the lack of data in the DHS datasets to measure them. Two additional outcome variables were studied, namely: the proportion of households using surface water for drinking and the percentage of households practising open defecation (Table 2). The DHSs provided information for each surveyed household on the primary type of facility used for drinking water, sanitation, and hand hygiene.

**Table 1. Features of the sampling design for the included DHSs [17–20].**

| Features of the sampling design | DHS-II | DHS-III | DHS-IV | DHS-V |
|---|---|---|---|---|
| Year | 2001 | 2006 | 2011–2012 | 2017–2018 |
| Number of strata | 13 | 23 | 23 | 23 |
| Number of PSUs drawn | 247 | 750 | 750 | 555 |
| Sampling frame of the UPS | RGPH-II | RGPH-III | RGPH-III | RGPH-IV |
| Number of households/PSUs drawn | 21–28 | 24 | 24 | 26 |
| Total number of expected households | 6,206 | 17,982 | 17,999 | 14,435 |
| Total number of households identified | 5,945 | 17,675 | 17,672 | 14,293 |
| Response rate (%) | 97 | 99 | 99 | 99 |
| Number of households surveyed (n) | 5,769 | 17,511 | 17,422 | 14,156 |

In addition, the following variables were extracted: i) characteristics of the household head: age (<30, 30–39, 40–49, 50–59, 60 and over), sex (male, female) and level of education (no formal education, primary, secondary, higher); ii) characteristics related to household composition and standard of living: household size (≤5, >5), existence of children under five in the household (yes, no), wealth index (poorest, poorer, middle, richer and richest) and iii) environmental characteristics: area (urban, rural) and department (Alibori, Atacora, Atlantic, Borgou, Collines, Couffo, Donga, Littoral, Mono, Ouémé, Plateau and Zou). Marital status of the household head and household wealth index variables were not available in DHS-II (2001).

## Data analysis

All analyses took into account the sampling designs of each survey. Household characteristics were described and compared between the surveys with the Chi-square test. For each survey, the indicators were calculated overall and according to household characteristics. Next, the Chi-square test permitted the investigation of the associations between household characteristics and the outcome variables. The study generated Annual Percent Changes (APCs) for each outcome variable, overall and between two successive surveys. The APC is

**Table 2. Standard definitions of outcome variables [9].**

| Components | Indicators | Definitions |
|---|---|---|
| **Water** | Proportion of households using basic drinking water services | Basic drinking-water services refer to improved water[1] points that provide drinking water provided that the round trip to collect water does not exceed 30 minutes, including waiting time. |
| | Proportion of households using surface water for drinking | Surface water is drinking water collected directly from a river, lagoon, dam, lake, pond, stream, canal or irrigation channel. |
| **Sanitation** | Proportion of households using basic sanitation services | Basic sanitation facilities are improved[2] facilities not shared with other households. |
| | Proportion of households practising open defecation | Open defecation is the disposal of human excreta in fields, forests, bushes, water bodies, beaches or other open spaces or with solid or liquid waste. |
| **Hygiene** | Proportion of households using basic hygiene services | These are facilities for washing hands with soap and water available in the home. |

[1] Piped supplies (tap water in the dwelling, yard or plot, public standposts) and non-piped supplies (bore-holes/tubewells, protected wells and springs, rainwater, packaged water, including, bottled water and sachet water, delivered water, including tanker trucks and small carts)

[2] Networked sanitation (flush and pour flush toilets connected to sewers) and on-site sanitation (flush and pour flush toilets or latrines connected to septic tanks or pits, ventilated improved pit la-trines, pit latrines with slabs, composting toilets, including twin pit latrines and container-based systems)

often used to measure trends in disease incidence and mortality rates [21–24]. The APC refers to the average increase or decrease expressed as a percentage of a given parameter between two successive units of time (usually the year) over a given period, assuming linearity. In the present study, the time unit considered was the year. A negative value indicates a decrease in the value of the parameter studied over the period considered, while a positive value suggests an increase.

Let $t_1$ to $t_2$ be a study period with $t_1$ the initial year, $t_2$ the final year, $P_{t_1}$ the value of a parameter $P$ at $t_1$ and $P_{t_2}$ the value of a parameter $P$ at $t_2$. The $APC$ of this parameter $P$ over the period $t_1$ to $t_2$ is given by the relation ($a$) [21]:

$$APC(\%) = \left[ \left( \frac{P_{t_2}}{P_{t_1}} \right)^{\frac{1}{t_2 - t_1}} - 1 \right] \times 100 \tag{a}$$

The APCs were disaggregated by household characteristics. In calculating the APCs, it was assumed that DHS-IV and DHS-V were conducted in the years "2011.5" and "2017.5" respectively, as these two surveys straddled two years, 2011–2012 and 2017–2018 respectively. Due to the absence of household head's marital status and household wealth index in DHS-II (2001), the overall APCs by these variables were calculated from 2006 to 2017–2018 instead of from 2001 to 2017–2018. Additionally, we previously emphasized that due to the former administrative division (Atacora, Atlantique, Borgou, Mono, Ouémé, Zou), in the DHS-II (2001), Atacora is referred to as Atacora and Donga; Atlantique, Atlantique and Littoral; Borgou, Borgou and Alibori; Mono, Mono and Couffo; Ouémé, Ouémé and Plateau; Zou, Zou and Collines. As a result, when calculating the APCs, household coverage of WASH services in Borgou (for example) for the year 2001 was considered individually for Borgou and Alibori. The same applies to Atacora and Donga, Atlantique and Littoral, Mono and Couffo, Ouémé and Plateau, Zou and Collines. To assess whether Benin is on track to reach the coverage target, assuming that trends over the past two decades remain unchanged, the following relationship was used ($b$) [22]:

$$P_{2030} = P_{2017-2018} \left( 1 + APC_{2017-2018} \right)^n \tag{b}$$

with $P_{2017-2018}$ the value of the parameter $P$ measured during DHS-V, $APC_{2017-2018}$ the APC calculated between 2001 and 2017–2018 and $n$ the number of years between 2017–2018 and 2030. Stata 15 and Excel were used to analyse the data.

## Ethical aspects

Launching data collection for the selected DHSs required ethical clearance from the relevant institutions. Depending on the survey, data collection was subject to ethical approval by the internal ethics committee of Macro International or ICF International. Additionally, given the survey, an ad hoc ethics committee established by the Ministry of Health or the National Committee for Ethics in Health Research also provided ethical authorization [17–20]. During data collection, eligible respondents sought informed consent before commencing interviews. The datasets used for the secondary analyses in this study were fully anonymised so that respondents could not be traced in any way. The ethical aspects of DHS-II, DHS-III, DHS-IV and DHS-V are detailed elsewhere [17–20].

## Results

### Basic household characteristics

S1 Table presents the basic characteristics of households. From 2001 to 2017–2018, household heads aged 30–39 were the most represented: 26.00% in 2001, 28.31% in 2006, 27.12% in

2011–2012 and 26.91% in 2017–2018. The percentage of female-headed households increased from 2001 to 2017–2018. Therefore, the male-to-female ratio among household heads decreased over the study period from 3.80 in 2001 to 3.02 in 2017–2018. The proportion of no formal-educated household heads declined (from 57.18% in 2001 to 52.64% in 2017–2018). In return, there was an increase in the share of household heads with higher education (from 2.83% to 5.58%). Generally, about 60% of households had children under the age of five. Overall, the households had five or fewer people (6 out of 10). Over the study period, about 4 out of 10 households lived in urban areas. In each survey from 2001 to 2017–2018, households in the Atlantique and Ouémé were the most represented.

## Level of access to WASH services

S2 Table shows the level of household access to drinking water, sanitation and hygiene in Benin. Household access to basic drinking water services was 50.54% (95% CI = 46.89–54.17) in 2001 and 63.98% (95% CI = 61.64–66.25) in 2017–2018 ($p<0.001$). The prevalence of surface water consumption among households was 5.84% (95% CI = 4.69–7.24) in 2017–2018, about half (10.54, 95% CI = 7.94–13.87) that observed in 2001 ($p<0.001$). S3–S7 Tables report the associations between household characteristics and the outcome variables. In general, there was less coverage of basic drinking water facilities among households headed by those aged 60 and over ($p<0.05$). Households led by female, higher educated or single individuals had significantly higher coverage of basic drinking water facilities and lower prevalence of surface water consumption ($p<0.05$). The same was true for richer/richest households, and those with five or fewer people, without children under five, in urban areas and in Littoral ($p<0.05$).

Household access to basic sanitation facilities was around 5.39% (95% CI = 4.60–6.32) in 2001 and increased to 13.29% (95% CI = 12.09–14.59) in 2017–2018 ($p<0.001$). Similar to the previous finding, coverage of basic sanitation facilities was globally higher in the richest households, and those with five or fewer people, without children under five, in urban areas and in Littoral ($p<0.05$). Also, households led by individuals aged 40–59 or more educated were associated with higher coverage of basic sanitation facilities ($p<0.05$). There was no significant difference according to the sex of the household head ($p>0.05$). In 2001, the prevalence of open defecation among households was 67.03% (95% CI = 63.84–70.07). In 2017–2018, it was still present in almost half (53.91, 95% CI = 51.33–56.46) of households. Open defecation was less common in households led by under-60, female, highly educated or single individuals ($p<0.05$). The same picture was held for the richest households, and those with five or fewer people, without children under five, in urban areas and in Littoral ($p<0.05$).

Household access to basic hygiene facilities was about 2% (95% CI = 1.68–2.68) in 2001 and about 10% (95% CI = 9.16–11.14) in 2017–2018 ($p<0.001$). In addition, disparities related to the level of education and marital status of the household head were found, as well as to the wealth index, household size, existence of children under five and location ($p<0.05$).

## APCs for access to WASH services

S8–S12 Tables display the APCs of household access to WASH services. Overall, access to basic drinking water facilities increased steadily by 1.44% each year (APC = +1.44%) from 2001 to 2017–2018 (from 50.54% to 63.98%). The highest APC (APC = +5.17%) was between 2001 and 2006. By household characteristics, the APC for coverage of basic drinking water facilities was highest for households in Littoral (APC = +3.80%), urban area (APC = +2.51%) and those headed by individuals with higher education (APC = +8.24%). The prevalence of surface water consumption fell from 10.54% to 5.84%, i.e. an APC of -3.52%, from 2001 to 2017–2018. Over this period, a decrease in surface water consumption among households was recorded at the

level of all groups. In particular, Zou had an APC = -8.50%. The national prevalence of surface water consumption decreased the most between 2006 and 2011–2012 (APC = -14.93%).

From 2001 to 2017–2018, the APC related to coverage of basic sanitation facilities was 5.62%. The highest APC was from 2006 to 2011–2012 (APC = +19.42%). Looking at their departments, Zou (APC = +20.22%), Donga (APC = +12.93%), Borgou (APC = +11.30%) and Atacora (APC = +11.29) had the highest APC. In addition, the richest households (APC = +10.83%), those in rural areas (APC = +10.93%) and headed by uneducated individuals (APC = +11.11%) had an APC related to coverage of basic sanitation facilities greater than ten percentage points from 2001 to 2017–2018. The prevalence of open defecation decreased from 67.03% to 53.91%, with an APC of -1.31%. This was mainly in Littoral (APC = -11.66%) and Zou (APC = -4.19%). The prevalence of open defecation decreased the most between 2006 and 2011–2012 (APC = -2.35%).

From 2001 to 2017–2018, the APC for coverage of basic hygiene facilities was +9.92%. The highest APC was between 2006 and 2011–2012 (APC = +31.70%). For households in rural areas, in Alibori, Collines and Borgou, with a low wealth index or headed by uneducated individuals, an APC of more than 20 percentage points was recorded.

### Projections of access to WASH by 2030

S13–S17 Tables set out projections of household access to WASH services. Notably, projections at the national level are presented in S1 and S2 Figs. If the trend between 2001 and 2017–2018 remains unchanged, the national coverage of basic drinking water facilities will rise to 76.50%, by 2030. Households headed by individuals with a higher level of education, in urban areas, in Littoral and Ouémé are on track for universal coverage. By 2030, the prevalence of surface water consumption is projected to be 3.73% nationally. Surface water consumption is expected to be eliminated in the richest and Littoral households, as well as in those headed by people with high levels of education.

If the trend from 2001 to 2017–2018 remains unchanged, national coverage of basic sanitation facilities is projected to be 26.33%, by 2030. Zou and Littoral will have the best coverage, with Zou on track for universal coverage. The same is true for the richest households. It is projected that in Alibori and Couffo, less than 10% of households will have these services. It also appears that less than one per cent of the poorest/poorer households will be able to access improved non-shared toilets. Nationally, the prevalence of open defecation is projected to be 45.71%, by 2030. The Littoral will continue to be the least (1.14%) affected department by open defecation.

According to projections, national coverage of basic hygiene services will be 32.98% in 2030. Borgou and Collines are on track for universal coverage. The same is true for the richer, middle, poorer and poorest households.

### Discussion

This work aimed to study the evolution of household access to adequate WASH services to determine the extent to which the commitments to the SDG targets are on track.

We highlight four key findings. First, Benin has made notable progress in household coverage of adequate WASH services over the past two decades, particularly between 2006 and 2011. Overall, access to basic drinking water facilities increased between 2001 and 2017–2018, from 50.54% to 63.98%. During the study period, household access to basic sanitation and hygiene services increased by a factor of 2.5 and 4.8, respectively. The prevalence of surface water consumption decreased by half, from 10.54% to 5.84%. Furthermore, the frequency of open defecation among households decreased by more than 13 percentage

points from 67.03% to 53.91%. This improvement may be linked to positive changes in socio-demographic and environmental factors associated with household access to adequate WASH services. These include the increased educational status of the population in general and household heads in particular, the enhancement of household wealth, the implementation of targeted interventions, the political commitment of decision-makers and the support of technical and financial partners. The level of education is an important social determinant of health that plays an essential role in the ability to make better decisions about one's health and increase the resources required to ensure the health of household members [25]. In Benin, the adult literacy rate increased by almost ten percentage points between 2002 and 2018 [26]. At the same time, this study found a decline in the proportion of non-educated household heads in favour of an increase in the percentage of household heads with higher levels of education, particularly from 2006 onwards. Therefore communication, information and awareness-raising interventions on the relevance of hygiene and basic sanitation were carried out in the general population and schools. From 2001 to 2017–2018, we can see an improvement in the conditions of people, with a poverty reduction, particularly between 2006 and 2011 [27]. During this period, the incidence of monetary poverty fell by 1.3 percentage points, from 37.5% to 36.2% [27]. As for non-monetary poverty, it affected 29.5% of individuals in 2011 against 44.1% in 2006, i.e. a drop of 14.6 percentage points, thus showing the improvement in household living conditions and, at the same time, the government's efforts to improve access to basic infrastructure (water, electricity) [27]. Some studies in Africa and Asia suggested that female-headed households were more likely to access adequate WASH services [28–30]. A study in Benin in 2022 showed results along these lines [11]. In the present study, we note that the proportion of female-headed households changed positively from 2001 to 2017–2018. Another point to note is United Nations Children's Fund (UNICEF)'s support in the fight against open defecation, particularly in rural areas, through the "Community-Led Total Sanitation" (CLTS) approach. From 2013 to 2017, the CLTS approach helped communities become aware of their conditions and commit to stopping open defecation [31]. From 2014 to 2017, 2,724 localities achieved end open defecation status [31]. At the strategic and policy level, there is a commitment by political authorities to promote people's access to WASH services. In the National Health Development Plan (PNDS) 2009–2017, the promotion of hygiene and basic sanitation was defined as a priority area, broken down into programmes whose efficient implementation would enable the health sector to meet the challenges [32]. The PNDS 2018–2022 also sets out an intervention axis promoting hygiene and basic sanitation [33]. Its goal 6 aims to guarantee access to water supply and sanitation services for all [33]. There has also been considerable investment to improve access to drinking water for the population, with the support of technical and financial partners. All these factors seem to have contributed to the increase in household coverage with adequate WASH facilities.

Secondly, there is a substantial difference in access to basic drinking water services compared to the sanitation and hygiene components. Although the percentage of households with access to basic sanitation and hygiene services significantly increased, during the period of interest, only 13.29% and 10.51% of households had access to these facilities in 2017–2018, respectively. One explanation lies in a lower commitment to sanitation and hygiene (in comparison to water). A 2018 situational analysis of the hygiene and basic sanitation sector indicated that the Government's Action Programme (2016–2021) did not prioritise the hygiene and basic sanitation sub-sector equal to the drinking water supply sub-sector [34]. In recent years, household surveys have adjusted the questions related to handwashing facilities to include responses for different types of handwashing facilities, including fixed devices (sinks or taps) and mobile devices (jugs, portable basins). Such was the case for DHS-V in 2017–2018.

The report shows that mobile facilities are in wide use in Benin [18]. Older surveys that do not include responses for mobile devices have potentially underestimated households with access to handwashing facilities. It may explain the low figures observed in 2001 and 2006.

Third, disparities related to demographic and socio-economic status groups remain. In each DHS (2001, 2006, 2011–2012 and 2017–2018), households with better coverage of basic WASH services belong to 'favourable' socio-economic groups. In 2022, a study using DHS-V found that the wealthiest households and few, and those headed by people aged 30 and over, female and with higher levels of education, were most likely to have access to basic WASH services [11]. This study shows that these disparities existed in the early 2000s and have continued over the past two decades. A better understanding of the factors associated with household access to basic WASH services is essential to identify the relevant interventions. For access to basic water sources, progress appeared to be stronger in households associated with favourable socio-economic conditions, raising concerns that inequalities may persist or even increase in the medium term. Projections suggest that if trends already observed continue, the poorest households and those living in rural areas will take the longest to achieve the goal of universal access to adequate water facilities. Regarding access to basic sanitation and hygiene facilities, the highest progress was among the lower socio-economic groups, partly because of the lack of coverage observed at the beginning of the study period. However, this progress was not yet sufficient to catch up with the better-off households in 2017–2018.

At the national level, projections suggest that Benin is not on track to reach universal coverage of basic water WASH services. If the trend from 2001 to 2017–2018 remains unchanged, national coverage of basic drinking water facilities is projected to be 76.50%. According to estimates, Sub-Saharan Africa will reach 75% coverage of basic drinking water facilities, which is in the same order of magnitude that we have projected for Benin [9]. Globally, projections suggest 94% coverage by 2030, which will not achieve universal access [9]. Only four regions have reached or are on track to achieve >99% coverage by 2030: Australia and New Zealand, Europe and North America, Latin America and the Caribbean, and East and South-East Asia [9]. Also, national coverage of basic sanitation facilities is projected to be 26.33% in Benin, which is similar to what has been projected for Sub-Saharan Africa (24%) [9]. However, on a global scale, 90% coverage will be reached by 2030 [9]. According to the present study, the prevalence of open defecation is projected at 45.71%. This frequency is still relatively high, especially since, assuming current rates of progress are maintained, most regions are on track to eliminate open defecation before 2030 [9]. We found that national coverage of basic hygiene services in Benin will be 32.98% by 2030. In Africa, at the current rate of progress, basic hygiene coverage would only reach 28% in 2030 compared to 78% globally [9].

There are some key strengths to this study. As far as we know, this study is the first to address this issue in Benin. The exploitation of a series of DHS datasets in Benin allowed the inclusion of large-size samples representative of the Beninese population at the national level and characterised by high response rates. Also, the use of comparable questionnaires in each survey contributed to the standardisation of the data collection process, making the data combination and comparisons relevant. The results of this study will enable national policymakers to understand the challenge of ensuring universal coverage of populations with adequate WASH services. Some limitations need to be pointed out. One is linked to the constraints of the variables available in the databases used, which did not allow for the identification of other factors of heterogeneity in the results found at the national level. Furthermore, the calculation of the APCs assumes linearity, which may not be verified. It could lead to an over- or underestimation of the projections made if the socio-economic, demographic, political and other contexts are more unfavourable or favourable than over the last twenty years.

## Conclusion

Benin has made significant progress in the coverage of households with adequate water, sanitation and hygiene services over the last two decades. A higher proportion of households had access to basic individual WASH services in 2017–2018 than in 2001, and a lower percentage of households consumed surface water and practised open defecation in 2017–2018 than in 2001. However, many more households had access to basic drinking water services than basic sanitation and hygiene services. Regarding the Sanitation and Hygiene components, a more significant commitment than that observed in previous years is needed. Disparities related to demographic and socio-economic status groups also remain. Furthermore, progress appears insufficient to meet the targets set in the Sustainable Development Goals. The strategies that have guided interventions implemented in recent years and currently will not guarantee universal coverage of households with basic WASH services by 2030. It is imperative to strengthen the ongoing strategies by targeting the most vulnerable groups and designing new ones based on evidence. It requires a better understanding of the drivers of inequalities in WASH coverage among populations. Analyses of solutions implemented in countries characterized by socio-economic contexts and other factors similar to those in Benin may be the subject of further work.

## Supporting information

**S1 Table. Basic household characteristics, Benin, 2001 to 2017–2018.**
(PDF)

**S2 Table. Level of household access to WASH services, Benin, 2001 to 2017–2018.**
(PDF)

**S3 Table. Association between household characteristics and access to basic drinking water services, Benin, 2001 to 2017–2018.**
(PDF)

**S4 Table. Association between household characteristics and surface water consumption, Benin, 2001 to 2017–2018.**
(PDF)

**S5 Table. Association between household characteristics and access to basic sanitation services, Benin, 2001 to 2017–2018.**
(PDF)

**S6 Table. Association between household characteristics and open defecation, Benin, 2001 to 2017–2018.**
(PDF)

**S7 Table. Association between household characteristics and access to basic hygiene services, Benin, 2001 to 2017–2018.**
(PDF)

**S8 Table. APCs of household access to basic drinking water services, Benin, 2001 to 2017–2018.**
(PDF)

**S9 Table. APCs of surface water consumption among households, Benin, 2001 to 2017–2018.**
(PDF)

**S10 Table. APCs of household access to basic sanitation services, Benin, 2001 to 2017–2018.**
(PDF)

**S11 Table. APCs of open defecation among households, Benin, 2001 to 2017–2018.**
(PDF)

**S12 Table. APCs of household access to basic hygiene services, Benin, 2001 to 2017–2018.**
(PDF)

**S13 Table. Projections of household access to basic drinking water services, Benin, 2019–2030.**
(PDF)

**S14 Table. Projections of household surface water consumption, Benin, 2019–2030.**
(PDF)

**S15 Table. Projections of household access to basic sanitation services, Benin, 2019–2030.**
(PDF)

**S16 Table. Projections of open defecation among households, Benin, 2019–2030.**
(PDF)

**S17 Table. Projections of household access to basic hygiene services, Benin, 2019–2030.**
(PDF)

**S1 Fig. Evolution of household access to individual basic WASH services from 2001 to 2017–2018, and projection to 2030, Benin.**
(PDF)

**S2 Fig. Evolution of household surface water consumption and open defecation from 2001 to 2017–2018, and projection to 2030, Benin.**
(PDF)

## Acknowledgments

We would like to thank the DHS programme for providing data for this study.

## Author contributions

**Conceptualization:** Nicolas Gaffan, Cyriaque Dégbey, Alphonse Kpozèhouen, Roger Salamon.

**Formal analysis:** Nicolas Gaffan.

**Methodology:** Nicolas Gaffan, Cyriaque Dégbey, Alphonse Kpozèhouen, Yolaine Glèlè Ahanhanzo.

**Supervision:** Roch Christian Johnson, Roger Salamon.

**Writing – original draft:** Nicolas Gaffan, Alphonse Kpozèhouen.

**Writing – review & editing:** Nicolas Gaffan, Cyriaque Dégbey, Yolaine Glèlè Ahanhanzo, Roch Christian Johnson, Roger Salamon.

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
