## [Decision Letter · Decision Letter 0]

9 Feb 2023

PONE-D-22-29273Is Benin on track to reach universal household coverage of basic water, sanitation and hygiene services by 2030?PLOS ONE

Dear Dr. GAFFAN,

Thank you for submitting your manuscript to PLOS ONE. After careful consideration, we feel that it has merit but does not fully meet PLOS ONE’s publication criteria as it currently stands. Therefore, we invite you to submit a revised version of the manuscript that addresses the points raised during the review process.

Both reviewers have provided some detailed and constructive comments which I think should be feasible to address but nevertheless require careful attention.

We look forward to receiving your revised manuscript.

Kind regards,

Alison Parker

Academic Editor

PLOS ONE

Journal Requirements:

2. 1. For studies reporting research involving human participants, PLOS ONE requires authors to confirm that this specific study was reviewed and approved by an institutional review board (ethics committee) before the study began. Please provide the specific name of the ethics committee/IRB that approved your study, or explain why you did not seek approval in this case.

Reviewers' comments:

Reviewer's Responses to Questions

**Comments to the Author**

1. Is the manuscript technically sound, and do the data support the conclusions?

Reviewer #1: Yes

Reviewer #2: Partly

2. Has the statistical analysis been performed appropriately and rigorously? 

Reviewer #1: Yes

Reviewer #2: Yes

3. Have the authors made all data underlying the findings in their manuscript fully available?

Reviewer #1: Yes

Reviewer #2: Yes

4. Is the manuscript presented in an intelligible fashion and written in standard English?

Reviewer #1: Yes

Reviewer #2: Yes

5. Review Comments to the Author

Reviewer #1: The authors endeavored to study this theme in Benin. The manuscript is very well prepared and the technical quality and presentation of the results is quite good. The results can be useful for decision makers in the country. I suggest that authors consider the modifications below for readability and further improvement of the article.

1. Line 68: what would 7c be?

2. In the third paragraph of the introduction, could you give examples of some vulnerable countries?

3. Write the abbreviation WASH in full in the introduction

4. What is the WASH estimated universal coverage rate for 2030?

5. Please, defines what DHS is.

6. Line 145: what were the missing variables?

7. Rewrite the phrase “The APC is the percentage …….assuming linearity” on line 201.

8. Review the numbering of equations in addition to naming them in the text.

9. Line 230-232: Do you know the reason for this decrease?

10. Wouldn't it be better to review the numbering of the tables? For example, Table 1 and on instead of Table S1?

11. It gets confusing to mention Tables S3-S7. I don't know if it's from table S3 to table S7 or just tables S3 and S7. Make this clear in your text.

12. Line 245-247: Enter the percentage obtained to aggregate the work.

13. "The prevalence of surface water consumption decreased the most between 2006 and 2011-2012 (APC = -14.93%)." Was that at the national or departmental level?

14. Define the acronym UNICEF and all others in the text.

15. I find the conclusion too short

16. Missing abbreviation list

17. I suggest improving the quality of the figures

18. Providing the link of each reference and mentioning the date of access is important.

Reviewer #2: In general, this paper is well-written, interesting and useful. This can be evidence for monitoring inequity. The paper is well-planned and easy to read.

The methodology appears sound, but I have made suggestions below that would improve or clarify the findings and some inputs for other parts (attached).

6. PLOS authors have the option to publish the peer review history of their article (what does this mean?). If published, this will include your full peer review and any attached files.

Reviewer #1: **Yes: **JOHNSON HERLICH ROSLEE MENSAH

Reviewer #2: No

---

## [Author Response · Author response to Decision Letter 0]

18 Mar 2023

Dear Editor,

We are grateful for the opportunity to submit a revised manuscript. We appreciate the efforts of the reviewers and the editorial team in providing suggestions for improving the quality of the manuscript. 

We have incorporated the changes into the revised manuscript. All authors have approved the current version of the manuscript.

Below are point-by-point responses to the comments.

Best regards.

GAFFAN Nicolas

[Corresponding author]

P.-S. - Line numbers in the responses are from the ‘Manuscript.docx’ file (without tracking changes).

Point-By-Point Responses

Reviewer 1

1. Line 68: what would 7c be?

“7c” is one of the targets of the Millennium Development Goals (MDGs). 

Lines 67-68: It called on states to "halve the proportion of people without sustainable access to safe drinking water and basic sanitation by 2015".

2. In the third paragraph of the introduction, could you give examples of some vulnerable countries?

In paragraph 3, we stated that SDG 1 includes a target to ensure universal access to basic services, especially for the poor and vulnerable. Addressing the most vulnerable implies attention to the specific WASH needs found in "special cases" such as refugee camps, detention centres, mass gatherings, pilgrimages, etc.

3. Write the abbreviation WASH in full in the introduction

Thank you for your comment. In the revised manuscript, we have defined the acronym WASH as the first use in the introduction. We did the same thing for the other abbreviations/acronyms.

Lines 103-105: Besides, estimates in 2020 showed that the world was on track to meet the 2030 Water, Sanitation and Hygiene (WASH) targets, with Sub-Saharan Africa and Oceania most off track.

4. What is the WASH estimated universal coverage rate for 2030?

Projections of people's access to WASH in 2030 globally and by region are available in the report "Progress on household drinking water, sanitation and hygiene 2000-2020: five years into the SDGs" by WHO/UNICEF Joint Monitoring Programme for Water Supply, Sanitation and Hygiene. According to estimates, 81% and 67% coverage of safely managed water and sanitation services are projected globally by 2030. Additionally, the world is expected to reach 94%, 90% and 78% coverage of basic WASH services by 2030, which will not achieve universal access. 

5. Please, defines what DHS is

Thank you for your comment. In the revised manuscript, we have defined DHS at the first use.

Line 130: DHS: Demographic and Health Survey

6. Line 145: what were the missing variables?

In the study, the level of access to basic hygiene services was assessed using two variables: the availability of hand hygiene facilities and the availability of water and soap. These last two variables are not available in the HOUSEHOLD dataset of the DHS-I in 1996.

7. Rewrite the phrase “The APC is the percentage … assuming linearity” on line 201.

Lines 195-197: We rewrite this as: “The APC refers to the average increase or decrease expressed as a percentage of a given parameter between two successive units of time (usually the year) over a given period, assuming linearity”.

8. Review the numbering of equations in addition to naming them in the text.

Thank you for your comment. The manuscript has been revised according to your suggestion. Lines 202 and 208: The first equation has been numbered (a) and the second (b). (a) and (b) were stated in the text.

9. Lines 230-232: Do you know the reason for this decrease?

There are several reasons for this finding: reinforcement of the application of laws promoting sex equality, strengthening the schooling of girls, increasing the rate of employment among women, and which gives them greater financial autonomy.

10. Wouldn't it be better to review the numbering of the tables? For example, Table 1 and on instead of Table S1?

Thank you for your comment. We have adopted this style of numbering as these tables have been presented in supplementary materials. The tables in the body of the manuscript were numbered: Table 1 and Table 2.

11. It gets confusing to mention Tables S3-S7. I don't know if it's from table S3 to table S7 or just tables S3 and S7. Make this clear in your text.

In the revised manuscript, we replace “Tables S3-S7” with “Tables S3 to S7” (line 239), and “Tables S8-S12” with “Tables S8 to S12” (line 263). 

12. Lines 245-247: Enter the percentage obtained to aggregate the work.

Thank you for your comment. In the revised manuscript, we have completed the percentages.

13. "The prevalence of surface water consumption decreased the most between 2006 and 2011-2012 (APC = -14.93%)." Was that at the national or departmental level?

It was at the national level.

14. Define the acronym UNICEF and all others in the text.

In the revised manuscript, we have defined UNICEF as the first use in the introduction. We did the same thing for the other acronyms

Line 337: UNICEF is United Nations Children’s Fund.

15. I find the conclusion too short

In the revised manuscript, the conclusion has been strengthened in response to the research question and policy implications (Lines 407-420).

16. Missing abbreviation list

Thank you for your suggestion. According to the instructions to the authors, the list of abbreviations is not required.

17. I suggest improving the quality of the figures

In the revised manuscript, the figures have been improved. In particular, we used the Preflight Analysis and Conversion Engine (PACE) digital diagnostic tool to ensure that the figures met PLOS requirements. We renamed the files "Figure 1.tif" and "Figure 2.tif" as "Figure S1.tif" and "Figure S2.tif", respectively.

18. Providing the link of each reference and mentioning the date of access is important.

Thank you for your suggestion. The list of references has been reviewed by the authors and omissions have been corrected where necessary.

Reviewer 2

1. Abstract 

The objective state 'make projection for 2030', but in conclusion, the Author has not clearly stated the projection result. The Author needs to add a statement to improve the conclusion. 

Thank you for your suggestion. There is a shortage of clarity in the answer to the research question in the conclusion. In the revised manuscript, the abstract has been strengthened in this sense (Lines 53-56).

2a. Methods

Sampling (line 151), I feel like there is missing information at ‘each department’. Add words/sentence that can connect with each department.

Lines 149-150: A rewording was made: “The departments were stratified into urban and rural, except for the Littoral, an entirely urban stratum”.

2b. Methods

In the introduction section, the authors have explained the differences in standard definitions of the WASH indicators in the MDGs and SDGs. The Author added reference information about the standard definition of WASH in table 1. 

Thank you for your comment.

2c. Methods

Table 1: The Author needs to add an explanation of how to create variables 'improved water' and 'improved sanitation' according to Benin DHS and references

Footnotes to Table 2: Following your suggestion, we have made some clarifying additions to the definition of an improved water source and an improved sanitation facility: 

• Improved water: Piped supplies (tap water in the dwelling, yard or plot, public standposts) and non-piped supplies (boreholes/tubewells, protected wells and springs, rainwater, packaged water, including, bottled water and sachet water, delivered water, including tanker trucks and small carts)

• Improved sanitation: Networked sanitation (flush and pour flush toilets connected to sewers) and on-site sanitation (flush and pour flush toilets or latrines connected to septic tanks or pits, ventilated improved pit la-trines, pit latrines with slabs, composting toilets, including twin pit latrines and container-based systems)

2d. Methods

Table 1 (lines 196-197): did this study conduct reanalyses following the operational definition written in table 1?

Table 1 provides details on the sampling for the selected demographic and health surveys. During the data analysis, the sampling design of the individual surveys was taken under consideration, via the 'svy' command in Stata 15.

2e. Methods

In lines 195-198, the Author needs to add information describing tables S1 and S2.

Line 223: Table S1 (supporting information) presents the basic characteristics of households.

Lines 235-236: Table S2 (supporting information) shows the level of household access to water, sanitation and hygiene in Benin. 

2f. Methods

Table S3, mention p value, is there a subgroup as reference?

Thank you for your suggestion. In Table S3, “p” is the p-value from the chi-square test of the association between household characteristics and access to basic water services (by survey). We have additional information in the footnote to facilitate the reading of the tables.

3a. Results

Benin DHS, which were carried out based on a sample with complex design. It is better to present the data displays with a 95% confidence interval. 

Thank you for your suggestion. For the data presented in Tables S1 to S7, the 95% confidence intervals were completed (supporting information). The 95% confidence intervals have also been added in the body of the manuscript.

3b. Results

In line 243, the Author needs to add the meaning of the p-value in tables S1 and S2.

Thank you for your suggestion. We have additional information in the footnote to facilitate the reading of the tables (Tables S1 to S7).

3c. Results

Line 264; although it has summarized tables S8-S12, it would be more communicative to add the word ‘Table S3’ before ‘the overall’ (line 265).

Thank you for your suggestion.

3d. Results

Line 270; what is the meaning of minus value? The Author needs to give a brief explanation.

A negative value indicates a decrease on average over the period considered, while a positive value suggests an increase. 

We were done in the "Data Analysis" subsection of the "Methods" section on how to interpret the APC. Line 197-199: A negative value indicates a decrease on average over the period considered, while a positive value suggests an increase. 

3e. Results

Projection of access to WASH by 2030, add number of table that is being discussed in the text of the result.

Thank you for your suggestion. We began the subsection with the sentence: Lines 288-289: “Tables S13 to S17 set out projections of household access to WASH services. Notably, projections at the national level are presented in Figures S1 and S2 (supporting information).”

4a. Discussion

Lines 311-315 are the justification of the study. I suggest these four sentences move to the end of the introduction. 

Thank you for your suggestion.

We argue for bringing it back to the end of the discussion just before the study's strengths and limitations (Lines 393-404).

4b. Discussion

Line 318, add '(Table S2)' after 63,68% to clarify this.

In the "Results" section, this result had already been presented by specifying Table 2. This is the proportion of households with access to basic water services. Line 235-237: “Table S2 (supporting information) shows the level of household access to water, sanitation and hygiene in Benin. Household access to basic water services was 50.54% in 2001 and 63.98% in 2017-2018”. 

4c. Discussion

Line 319-320 – which table was referred to? State the number of tables

It is Table S2 (supporting information).

The percentage of households with basic sanitation services increased 2.5 times, from 5.39% to 13.29% between 2001 and 2017-2018 (lines 246-247). The percentage of households with basic hygiene services increased 4.8 times, from 2.12% to 10.11% between 2001 and 2017-2018 (Lines 257-258).

4d. Discussion

The SDGs' slogan is no one leave behind and interconnection between SDGs’ goals. This paper will be interesting if it discusses connection with the goal#10- inequity that affects the achievement of the 2030 target in Benin. This paper can be evidence-based for monitoring inequity in Benin when discussing the characteristics of being able to raise which subgroups are disadvantaged so that interventions can address these sub groups with the proper programs to reduce the gap between subgroups. Then these will increase the achievement of WASH targets.

Thank you for your suggestion. It will certainly be studied in the future. Goal 10 will not be addressed here.

4e. Discussion

The Author can add discussion abour policy recommendations for Benin's government based on this study.

In the revised manuscript, in the conclusion, we have added policy recommendations for the government of Benin (Lines 407-420).

5. Conclusion

Rewrite more clearly to answer the research question related to the projection result.

Thank you for your suggestion. In the revised manuscript, the conclusion has been strengthened in this sense.

6a. Tables; make the title of the table and text. Example text: 'Level of household access to water' title of table S3: 'access to basic drinking water services'. 

Thank you for your suggestion. The authors have reviewed the titles of all tables and modified them where necessary.

6b. Table presentation – to be self-reported. Add a footnote about the p-value – what you want to explain from the p-value so that the table is easier to read.

Thank you for your suggestion. We have additional information in the footnote to facilitate the reading of the tables

6c. In each table, add in lowest row for total Benin, ‘n’ and ‘%’ in each column.

Thank you for your suggestion. Under your comment, we have added a final line that shows the total for Benin.

6d. In table S1-S7, it is better to add the year after DHS II ‘(2001)’, DHS III ‘(2006)’, DHS IV ‘(2011-2012)’, DHS V ‘(2017-2018)’

Thank you for your suggestion. In the individual tables, the survey years have been added.

Other comments

1. For studies reporting research involving human participants, PLOS ONE requires authors to confirm that this specific study was reviewed and approved by an institutional review board (ethics committee) before the study began. Please provide the specific name of the ethics committee/IRB that approved your study, or explain why you did not seek approval in this case

The study included DHS-II, DHS-III, DHS-IV, and DHS-V in Benin. Depending on the survey, data collection was subject to ethical approval by the internal ethics committee of Macro International or ICF International. Additionally, depending on the survey, an ad hoc ethics committee established by the Ministry of Health or the National Committee for Ethics in Health Research also provided ethical authorization.

2. We note your current Data Availability statement is: "No - some restrictions will apply" "Data are available after a request sent via the DHS Program website at https://dhsprogram.com/."

The dataset we have used is available free of charge, upon request, and without restriction on the DHS program website (‘https://dhsprogram.com/data/available-datasets.cfm’).

In addition to the above comments, all spelling and grammatical errors have been corrected. Also, minor inputs have been made to improve the quality of the work. We have placed all the supplementary materials in a .zip file.

We hope that the esteemed reviewers will be able to see that the revised manuscript has taken constructive criticism into account.

---

## [Decision Letter · Decision Letter 1]

10 May 2023

Is Benin on track to reach universal household coverage of basic water, sanitation and hygiene services by 2030?

PONE-D-22-29273R1

Dear Dr. GAFFAN,

We’re pleased to inform you that your manuscript has been judged scientifically suitable for publication and will be formally accepted for publication once it meets all outstanding technical requirements.

Kind regards,

Alison Parker

Academic Editor

PLOS ONE

Additional Editor Comments (optional):

Reviewers' comments:

Reviewer's Responses to Questions

**Comments to the Author**

1. If the authors have adequately addressed your comments raised in a previous round of review and you feel that this manuscript is now acceptable for publication, you may indicate that here to bypass the “Comments to the Author” section, enter your conflict of interest statement in the “Confidential to Editor” section, and submit your "Accept" recommendation.

Reviewer #1: All comments have been addressed

Reviewer #2: All comments have been addressed

2. Is the manuscript technically sound, and do the data support the conclusions?

Reviewer #1: Yes

Reviewer #2: Yes

3. Has the statistical analysis been performed appropriately and rigorously? 

Reviewer #1: Yes

Reviewer #2: Yes

4. Have the authors made all data underlying the findings in their manuscript fully available?

Reviewer #1: Yes

Reviewer #2: Yes

5. Is the manuscript presented in an intelligible fashion and written in standard English?

Reviewer #1: Yes

Reviewer #2: Yes

6. Review Comments to the Author

Reviewer #1: (No Response)

Reviewer #2: The Author has already revised the first draft paper based on the reviewer’s inputs.

In general, paper is better sound, and the tables are self-reported with footnote.

But I have 2 inputs for conclusion and note (in supporting information file).

Conclusion (line 410-423):

The conclusion has been revised but according to it to be too long. Some parts are more appropriate to be included in the method or become a footnote in relevant tables. The author needs to revise again the conclusion to be more concise and answer the objectives of this study. Whether the Benin projection results are on the tract in line with the 2030 target or still far from target. Recommendations should be more operational and addressed to whom? Which departments are progressing, and which departments need more attention and effort to support the achievement of Benin 2030 and for which indicator?

Note (in supporting information file):

This note will be more concise when included in the methods, if there is no word count restriction.

7. PLOS authors have the option to publish the peer review history of their article (what does this mean?). If published, this will include your full peer review and any attached files.

Reviewer #1: No

Reviewer #2: No

---

## [Editor Report · Acceptance letter]

18 May 2023

PONE-D-22-29273R1 

Is Benin on track to reach universal household coverage of basic water, sanitation and hygiene services by 2030? 

Dear Dr. GAFFAN:

I'm pleased to inform you that your manuscript has been deemed suitable for publication in PLOS ONE. Congratulations! Your manuscript is now with our production department. 

Kind regards, 

on behalf of

Dr. Alison Parker 

Academic Editor

PLOS ONE